# Experimental Data of a Hexagonal Floating Structure under Waves

Roman Gabl [1],* , Robert Klar [2],* , Thomas Davey [1] and David M. Ingram [1]

1   School of Engineering, Institute for Energy Systems, FloWave Ocean Energy Research Facility,
    The University of Edinburgh, Max Born Crescent, Edinburgh EH9 3BF, UK;
    tom.davey@flowave.ed.ac.uk (T.D.); David.Ingram@ed.ac.uk (D.M.I.)
2   Unit of Hydraulic Engineering, University of Innsbruck, Technikerstraße 13, 6020 Innsbruck, Austria
*   Correspondence: roman.gabl@ed.ac.uk (R.G.); robert.klar@uibk.ac.at (R.K.)

**Abstract:** Floating structures have a wide range of application and shapes. This experimental investigations observes a hexagonal floating structure under wave conditions for three different draft configurations. Regular waves as well as a range of white noise tests were conducted to quantify the response amplitude operator (RAO). Further irregular waves focused on the survivability of the floating structure. The presented dataset includes wave gauge data as well as a six degree of freedom motion measurement to quantify the response only restricted by a soft mooring system. Additional analysis include the measurement of the mass properties of the individual configuration, natural frequency of the mooring system as well as the comparison between requested and measured wave heights. This allows us to use the provided dataset as a validation experiment.

**Keywords:** floating structure; hexagonal; validation experiment; wave; motion capturing; white noise; regular and irregular waves; motion capturing

## 1. Introduction

The presented experimental investigation is part of an ongoing research project to develop the concept of the Buoyant Energy Quarters (BEQ) [1]. This floating structure combines electrical storage capability based the pumped-storage hydropower [2,3] with the possibility to provide additional highly needed living space for cities close to the coast. The key part of the investigated design is a standardised of modular approach based on a hexagonal shape. This basic geometry can be use as a validation experiment to ensure that further adaptations including the design of the mooring system can be conducted based on numerical simulations. Hence, a specific validation experiment was conducted in the FloWave Ocean Energy Research Facility to investigate a scaled version of the device in the wave tank with the focus on the response amplitude operator (RAO) as well as the survivability of the structure. This paper describes the experimental set-up and introduces the available data set [4], which can be use as an independent validation experiment.

The intended floating structures will be a substantial expansion of a city and can be classified as a very large floating structure (VLFS). Simulations of such VLFS are typically conducted under the assumption of small relative motion, which allows the usage of solvers in the frequency domain [5,6]. Successful validation experiments for rectangular floating structures can be found in Yoon and Cho [7] as well as for VLFS with inner water levels [8]. Karperaki and Belibassakis [9] presented a method to include the interaction of the seabed bathymetry based on a 2D-approach and Wang and Gu [10] investigate the specific design

challenges of VLFS. A combination with a wave energy converter (WEC) can be found by Nguyen and Wang [11] and Thomsen et al. [12,13].

Different concepts of floating solutions are proposed in combination with wind energy production [14,15]. This introduces specific challenges related to conflicting scaling laws, which have to be considered in the experimental investigation [16]. The connection of a large number of floating wind devices [17] or floating photovoltaic plants [18,19] faces comparable challenges and a high degree of modularity is a key goal of the design. Using more traditional geometry, Subramanian et al. [20] investigates a suction stabilized floating platform. Floating structures can also be used to support bridges [21,22], as breakwaters combined with a WEC [23] or as a multi purpose platform [24,25].

Numerical simulations are capable of modelling and describing even complex phenomena [26]. Each individual investigation has to be verified to ensure a high quality result. Furthermore, validation experiments allow a direct comparison of the numerical results with measurements conducted in nature or the laboratory. Typically those experiments are conducted with a very simple geometry, which can be standardised, to investigate a specific phenomena with as few variables as is practical. For example, Kramer et al. [27] investigated a hollow sphere and its heave response dropping into the water and a simplified cylindrical buoy was deployed for the mooring investigations presented by Jiang et al. [28]. Gabl et al. [29,30] used a hollow cylinder to compare the effects of solid and water ballast. It was observed that differences caused by the sloshing occurred in combination with big motions, which supports the chosen simplification of this presented experiment. As later described, the mass properties of the architecture and the stored different inner water levels were combined to one overall mass distribution. The three investigated drafts represents a broad range of filling levels as well as additional usage of the floating structure.

A detailed description of the experimental set-up including the used measurement instrumentation as well as the wave conditions can be found in Section 2. Further detailed analysis are provided in Section 3, which include the mass properties of the three different drafts, the natural frequency of the used mooring system as well as the actual measured wave amplitude/height. The final Section 4 describes the available dataset, which can be accessed via Edinburgh DataShare [4] to be used a validation experiment of a floating hexagonal structure.

## 2. Experimental Set-Up

### 2.1. Floating Object

The floating object was investigated in FloWave Ocean Energy Research Facility, which is a wave and current test facility mainly targeted at renewable energy devices [31–36]. For the presented investigation only the 360° wave capability was used. The diameter of the round tank is 25 m and the upper part has a constant water depth of 2 m. Under the 1 m thick floor construction (inner part is raiseable to allow dry access) is a similar water volume, which acts as a recirculating chamber to provide a current speed of over 1.6 m/s from any direction generated with twenty-eight flow-drive units.

The presented investigation used a hollow hexagonal prism for the basic outside structure. For the definition of the volume the side length $s$ with 280 mm and the total height $H$ of 289 mm is needed. It is split into a bottom part including a ground plate as well as the six vertical walls, which are all made of 12 mm PVC plates. The outside shape is identical for all three investigated configurations. As presented in Table 1, the weight and mass properties were varied by adding different inside structures, which not only simulate different filling levels but also include the influence of potential architectural structures on top of the floating storage solution. Figure 1 shows the configuration $D$1 with the largest draft of the three investigated cases.

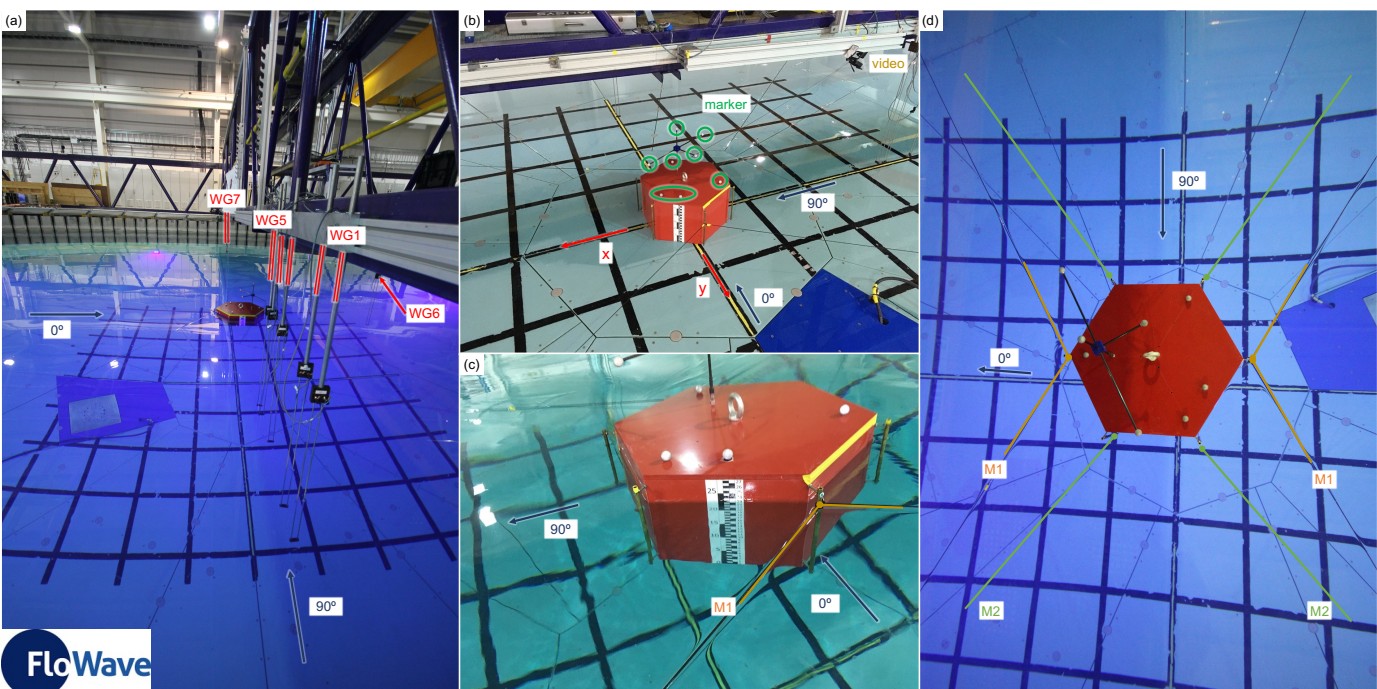

**Figure 1.** Overview of the experimental set-up for the configuration D1: (**a**) wave gauges (WG) and definition of the wave direction in the tank coordinate system. (**b**) model on the raised floor and highlighted markers for the motion capturing system (**c**) model under wave conditions coming from 0°. (**d**) top view to visualise the two different mooring options *M*1 and *M*2.

**Table 1.** Properties of the three different investigated configurations including the specific draft $h$ (vertical distance from the bottom of the floating structure to the still water level (WL) of the tank) in relation to the total height $H$ of the prism as well as vertical distance $h_{CG,B}$ from the bottom to the centre of gravity–theoretical mass properties: distance between the water level and the centre of gravity $\Delta h_{CG,WL}$ and the moment of inertia $I_{x,WL}$ and $I_{y,WL}$ with a reference axis at the height of the WL as well as $I_z$.

| Units | Weight [kg] | Draft $h$ [mm] | $h/H$ [%] | $h_{CG,B}$ [mm] | $\Delta h_{CG,WL}$ [mm] | $I_{x,WL} = I_{y,WL}$ [kg m$^2$] | $I_z$ [kg m$^2$] |
|---|---|---|---|---|---|---|---|
| Conf $D1$ | 47.70 | 234 | 81.0% | 120 | −114 | 1.789 | 1.620 |
| Conf $D2$ | 36.13 | 177 | 61.4% | 113 | −64 | 1.111 | 1.322 |
| Conf $D3$ | 24.31 | 119 | 41.3% | 121 | 1 | 0.780 | 0.944 |

Two main direction for the waves can be defined due to the hexagonal base plate. The first one is aligned with two opposing corners and the other is perpendicular to a side wall. Those wave directions are identical for steps in 60° around the model and represent the occurring extreme values. The wave tank FloWave allows to investigate waves from any direction hence it is encircled by 168 absorbing wave makers. Nevertheless, the investigation was limited to the two previously mentioned main directions. It is assumed that those directions result in the largest motion response and that all other direction lead to an additional rotation around the *z*-axis. The exact values of rotation is depending on the used very soft mooring system. Consequently, a precise modelling of the mooring system is required, which is not beneficial for a validations experiment hence another uncertainty is introduced in the comparison. In case of an actual deployment of such a structure, this rotation would be restricted by the mooring system. It was assumed that the incoming wave, which was orientated perpendicular to one side of the hexagonal structure, result in the largest run up on the side of the floating object. This was selected as the main wave direction for this presented investigation. The moveable gantry allows access to the tank and a simple installation of wave gauges (WG) in a line. Consequently, this direction was chosen as the main wave direction and aligned with the *x*-axis for the

local coordinate system. In the tank definition, which splits the full 360° in two half from 0° to ±180° (the boundaries are identical), this is the wave direction of 90°. The second main direction of the hexagonal geometry is tested with waves based on a 0° direction (Figure 1). The origin of the tank coordinate system is the centre of the tank, where the model was initially placed. The right handed coordinate system has an upright *z*-axis.

Figure 1d shows a top view of the experimental set-up with the two main wave directions 0° and 90° and the two different options for the connection of the mooring lines are highlighted. Both mooring systems use the metal attachment rails, which were added at all corners of the floating structure. This allows the connection of the mooring lines at the still water surface. A combination between a stiff line and a hollow elastic of 3 m long (diameter 3 mm) was used. Each line was connected to the edge protection of the tank at the quadrants points (rotated by 45°) and introduce a pre-tension. This allows a good station keeping without introducing to many restraints, which was beneficial for a wide range of previous experiments including floating objects [29,30,37]. Both mooring configurations were symmetrical around the *x*-axis and had the identical attachment points tank side. Two lines are joined together at a single point for the mooring variation *M*1. This introduces a minimum restraint for the rotation around the *y*-axis, which is the main rotation for the wave direction of 90°. Each individual line is connected to a separate edge for the second configuration *M*2 causing a increased influence in relation to the *M*1. All irregular wave tests including the JONSWAP spectra as well as white noise investigations were only conducted with the mooring configuration *M*1. The description of the used instrumentation is provided in Section 2.2 and the investigated wave cases are summarised in Section 2.3.

## 2.2. Instrumentation

The experimental set-up includes two main measurement systems recording with the identical measurement frequency of 128 Hz: (a) wave gauges (WG) and (b) Motion capturing system (MoCAP). The measurement of the free surface elevation was conducted with seven conductive WG. Each was calibrated daily in the range of ±100 mm in five steps. This process provides a high accuracy capture of the local free surface elevation, which is typically smaller than 1 mm [38,39]. Six of the seven WG were placed in the main symmetry plane (along the *x*-axis, with *y* = 0 m). Only WG 6 is placed with an offset of 1.68 m in the negative *y*-direction (on the other side of the gantry) to provide in phase measurements for the floating structure, which was placed at the tank centre. WG1 to WG5 represents a fixed reflection array based on a Golumb ruler with a total length of 1 m. The marks defining the spacing are also include in the overview in Table 2. The WG5 and WG7 were place in a distance of 6 or 12 times, respectively, the side length of the hexagon (*s* = 280 mm). This wide range was chosen to ensure that the floating model did not collide with the WG.

**Table 2.** Location of the seven wave gauges (WG) in the tank coordinate system shown in Figure 1.

|  | **WG1** | **WG2** | **WG3** | **WG4** | **WG5** | **WG6** | **WG7** |
|---|---|---|---|---|---|---|---|
| *x* [m] Model | −2.68 | −2.59 | −2.32 | −1.86 | −1.68 =−6 × 0.28 [m] | 0.00 model | 3.36 =12 × 0.28 [m] |
| Golomb ruler | 0 | 1 | 4 | 9 | 11 | | |
| *y* [m] | 0 | 0 | 0 | 0 | 0 | −1.65 | 0 |

As highlighted in Figure 1b, eight markers were placed on the floating structure, each observed by the MoCAP system provided by Qualisys (Göteborg, Sweden). The 2D-view of all eight cameras was combined in the software Qualisys Track Manager (QTM, version 2020.3, Qualisys, Göteborg, Sweden) delivering the 3D location of each marker in relation to the global coordinate system (identical to the tank coordinate system shown in Figure 1). Based on those individual markers, a rigid body definition was used to calculate the six degree of freedom (DoF) of the floating structure. Four markers were place on the lid

of the floating box and used to define this local body definition. The axis were chosen similar to the global coordinate system and the origin was always set at the still water surface height. Consequently, each investigated configuration has a slightly different body definition compensating the change in the draft. A further four markers are placed in a tree structure to improve the overall capture quality. Refinement calibrations were conducted each day including an additional one in the afternoon caused to account for varying lighting conditions. The body residuals were always smaller 1 mm.

Both measurement systems were synchronised with a digital pulse send by the wave makers. The measurement frequency for all system was set to 128 Hz, which allows a very detailed observation.

### 2.3. Wave Conditions

Three different types of waves were investigated as part of this study: (a) regular waves (Table 3), (b) white noise (Table 4) and (c) JONSWAP irregular waves (Table 5).

**Table 3.** Wave conditions for the regular wave cases including the two mooring cases *M*1 and *M*2 (Figure 1d) and wave direction. The requested wave amplitude $a_W$ was set to a constant 5 mm–Capture time 170 s, run time 160 s and repeat time of 128 s.

| Mooring Conf.→ | | M1 | M1 | M2 | M2 | M1 | M1 | M2 | M2 | M1 | M1 | M2 | M2 | - | - |
| Wave Angle [°]→ | | 0 | 90 | 0 | 90 | 0 | 90 | 0 | 90 | 0 | 90 | 0 | 90 | 0 | 90 |
| Wave | $f_W$ [Hz] | Conf D1 | | | | Conf D2 | | | | Conf D3 | | | | Empty Tank | |
|---|---|---|---|---|---|---|---|---|---|---|---|---|---|---|---|
| Reg1 | 0.3 | x | x | x | x | x | x | | | x | x | x | x | x | x |
| Reg2 | 0.4 | x | x | x | x | x | x | | | x | x | x | x | x | x |
| Reg3 | 0.5 | x | x | x | x | x | x | | | x | x | x | x | x | x |
| Reg4 | 0.6 | x | x | x | x | x | x | | | x | x | x | x | x | x |
| Reg5 | 0.65 | x | x | x | x | x | x | | | x | x | x | x | x | x |
| Reg6 | 0.7 | x | x | x | x | x | x | | | x | x | x | x | x | x |
| Reg7 | 0.725 | x | x | x | x | x | x | | | x | x | x | x | x | x |
| Reg8 | 0.75 | x | x | x | x | x | x | | | x | x | x | x | x | x |
| Reg9 | 0.775 | x | x | x | x | x | x | | | x | x | x | x | x | x |
| Reg10 | 0.8 | x | x | x | x | x | x | | | x | x | x | x | x | x |
| Reg11 | 0.825 | x | x | x | x | x | x | | | x | x | x | x | x | x |
| Reg12 | 0.85 | x | x | x | x | x | x | | | x | x | x | x | x | x |
| Reg13 | 0.875 | x | x | x | x | x | x | | | x | x | x | x | x | x |
| Reg14 | 0.9 | x | x | x | x | x | x | | | x | x | x | x | x | x |
| Reg15 | 0.925 | x | x | x | x | x | x | | | x | x | x | x | x | x |
| Reg16 | 0.95 | x | x | x | x | x | x | | | x | x | x | x | x | x |
| Reg17 | 0.975 | x | x | x | x | x | x | | | x | x | x | x | x | x |
| Reg18 | 1 | x | x | x | x | x | x | | | x | x | x | x | x | x |
| Reg19 | 0.8125 | x | x | x | x | x | x | | | x | x | | | x | x |
| Reg20 | 0.8375 | x | x | x | x | x | x | | | x | x | | | x | x |
| Reg21 | 0.85 | x | x | x | x | x | x | | | x | x | | | x | x |
| Reg22 | 0.8625 | x | x | x | x | x | x | | | x | x | | | x | x |
| Reg23 | 0.8875 | x | x | x | x | x | x | | | x | x | | | x | x |
| Reg24 | 0.7625 | x | x | x | x | x | x | | | x | x | | | x | x |
| Reg25 | 0.7875 | x | x | x | x | x | x | | | x | x | | | x | x |
| Reg26 | 0.45 | x | x | x | x | x | x | | | x | x | | | x | x |
| Reg27 | 0.425 | x | x | x | x | x | x | | | x | x | | | x | x |
| Reg28 | 0.85 | x | x | x | x | x | x | | | x | x | | | x | x |
| Reg29 | 0.675 | | | | | x | x | | | x | x | | | x | x |
| Reg30 | 0.7125 | | | | | x | x | | | x | x | | | x | x |
| Reg31 | 0.7 | | | | | x | x | | | x | x | | | x | x |
| Reg32 | 0.625 | | | | | | | | | x | x | | | x | x |
| Reg33 | 0.6625 | | | | | | | | | x | x | | | x | x |

Capture, run, and repeat times are defined for each wave. The repeat time is part of the fundamental wave definition and it ensures that a full number of wave repeats have to be fit into the repeat time, which can result in a difference between the requested and generated wave frequencies [40]. At the start and the end of this period the waves are identical. The run and capture time start with the activation of the wave makers. While the run time only covers the time the wave maker were active, the capture time for all the instrumentation is set longer to document the remaining waves in the tank. Further details are presented in Section 4.

A key finding of the investigation was the quantification of the response amplitude operator (RAO) for all 6 DoF. Therefore, a frequency sweep of regular waves were run covering wave frequencies $f_W$ from 0.3 to 1 Hz and a constant requested wave amplitude of 5 mm. An initial set of waves was defined (Reg1 to Reg18), which was refined further depending on the preliminary analysis of the RAO.

The starting point was the model configuration $D1$ (largest draft) and mooring $M1$. The same waves were repeated for the $M2$ version as well as for the configuration $D2$. Additional frequencies were again added but $M2$ was not investigated due to time constraints. A similar process was conducted for the model $D3$ but in this case only the first 18 waves were conducted with the second mooring configurations. The full set of waves were repeated in the absence of the model (open tank). A summary of the chosen waves are presented in Table 3.

As an alternative to monochromatic regular wave testing, the RAOs can be calculated from a polychromatic random noise tests. This includes pink noise [41] but most commonly white noise (WN) waves are used for the investigation of floating structures [42–45]. This irregular wave covers a randomly a frequency band by maintaining a constant power spectral density. This irregular wave is defined by the requested wave height $H_{m0,W}$, wave direction as well as a minimum and maximum wave frequency $f_W$. Different combinations of those parameters were chosen to compare the results against the method based on multiple single regular waves. Table 4 presents the investigated combinations. The WN tests were conducted for all three model variations $D1$ to $D3$ based on the mooring configuration $M1$ and only for $D1$ also for $M2$. An open tank measurement was also added for all WN waves.

**Table 4.** Wave conditions for the white noise with a variable requested wave height $H_{m0,W}$ and minimum and maximum wave frequency band as well as a constant random seed of 3.

| Wave | $H_{m0,W}$ [m] | Frequency min [Hz] | max [Hz] | Wave Direction [°] | Capture [sec] | Run Time [sec] | Repeat [sec] |
|------|------|------|------|------|------|------|------|
| WN1 | 0.01 | 0.2 | 0.8 | 90 | 170 | 160 | 128 |
| WN2 | 0.01 | 0.2 | 0.9 | 90 | 170 | 160 | 128 |
| WN3 | 0.005 | 0.2 | 1 | 90 | 170 | 160 | 128 |
| WN4 | 0.005 | 0.5 | 1.1 | 90 | 170 | 160 | 128 |
| WN5 | 0.01 | 0.5 | 1.1 | 90 | 170 | 160 | 128 |
| WN6 | 0.015 | 0.5 | 1.1 | 90 | 170 | 160 | 128 |
| WN7 | 0.0025 | 0.5 | 1.1 | 90 | 170 | 160 | 128 |
| WN8 | 0.005 | 0.2 | 1.1 | 90 | 170 | 160 | 128 |
| WN9 | 0.005 | 0.2 | 1.1 | 0 | 170 | 160 | 128 |
| WN10 | 0.005 | 0.2 | 1.1 | 90 | 550 | 540 | 512 |
| WN11 | 0.005 | 0.2 | 1.1 | 0 | 550 | 540 | 512 |

The regular waves and the white noise test provide the motion RAO of the different floating structures and can be used as a validation experiment for future numerical comparisons. The stability against extreme wave conditions were investigated based on an additional set of irregular waves (JONSWAP), which are presented in Table 5. Only the requested wave height $H_{m0,W}$ is changed up to a point when the waves over-top the floating structure. This was reached with a $H_{m0,W}$ of 0.03 m for the configuration with the

largest draft *D*1 and consequently smallest freeboard. This wave was repeated and also run with a wave direction of 0°. The other two configurations were tested with the same waves and a $H_{m0,W}$ of 0.03 m and two additional increased wave heights. This investigation was limited to the mooring configuration *M*1. All waves were also measured as an open tank experiment.

**Table 5.** Wave conditions for the JONSWAP irregular (Irr) wave cases including variable requested wave height $H_{m0,W}$ depending on the specific model configuration as well as a constant wave direction of 90°, gamma value of 3.3 and period of 1.18 s (=1/0.85 Hz)–Capture time 1540 s, run time 1530 s and repeat time of 1500 s

| $H_{m0,W}$ [m] | Wave Direction [°] | Conf D1 | Conf D2 | Conf D3 | Empty Tank |
|---|---|---|---|---|---|
| 0.01 | 90 | Irr1 | | | Irr1 |
| 0.02 | 90 | Irr2 | | | Irr2 |
| 0.03 | 90 | Irr3 and Irr4 | Irr1 | Irr1 | Irr3 |
| 0.03 | 0 | Irr5 | Irr2 | Irr2 | Irr5 |
| 0.04 | 90 | | Irr3 | Irr3 | Irr4 |
| 0.06 | 90 | | Irr4 | Irr4 | Irr6 |

## 3. Additional Analysis

The following section presents three additional analyses, which improve the usability of the presented dataset. Section 3.1 focuses on the floating device itself and compares the theoretical mass properties with the measured values. The natural frequency of the mooring system is presented in Section 3.2 and a comparison of the requested with the actual measured wave amplitudes are provided in Section 3.3.

### 3.1. Mass Properties of the Model

The presented experimental investigation is intended to be used as a validation experiment for numerical simulations, which will be used as the main tool for the further optimisation of the floating platform. An important first step was to ensure that the theoretical mass distribution (Table 1) for each of the three variations was realised and also provide a confidence interval for each value.

The external shape of the model was fixed and the changes were implemented with different ballast option as inputs. One of them is shown in Figure 2c. It had a core plate made of concrete combined with plastic plates and cans filled with wolfram balls. The precise distance from the inside bottom was adjusted with three threaded rods. After the assembly, each version of the model was investigated in the trimming tank in FloWave (Figure 2b) to ensure that the model floated correctly. The mass properties, namely the centre of gravity (CG) and moment of inertia (MI), were measured with a swing acting as a pendulum [46]. Figure 2a presents a picture of the model for the static investigations. Therefore, additional calibration weights were added and the angle of the new equilibrium was measured. In addition, a dynamic measurement was conducted based on the assumption of a pendulum to evaluate the moment of inertia in relation to the pivot axis. The values for the empty swing as well as further information can be found in Gabl et al. [47].

The centre of gravity (CG) was measured for both pivot axes with each two different distances of the additional calibration weights. Table 6 presents the results for each axes as well as a mean value. The difference to the theoretical value is relatively small, especially for the lower pivot axis *P*1. Larger differences were found for the higher pivot axis *P*2. It can be assumed that the theoretical values could be reached very accurately and the potential range for a sensitivity analysis of the numerical simulation for this value is approximately ±5 mm.

The maximum difference of the measured to the theoretical value of the CG occurred for the pivot axis *P*2. Hence the CG of the swing is very close to the lower pivot axis *P*1, the swing itself has a bigger influence on the measurement for the higher pivot point.

In addition, the weight of the swing is 42.28 kg, which is only slightly lower than the heaviest version of the model. This was also a potential reason why the results for the moment of inertia (MI) based on the measurements conducted with the pivot point $P2$ showed greater difference to the theoretical value in Table 7. This table also provide the MI for the model and the swing around the same axis, which were similar to the individual water level. $I_{Mod+Swing,WL}$ was in the range of 60 kg m$^2$ and the model only added around 1 kg m$^2$ to this sum. Consequently, it has to be highlighted that the measurement had some limitations and there was not an ideal ratio between swing and model. The confidence range is in the range of 20% of the MI due to the measurement uncertainties.

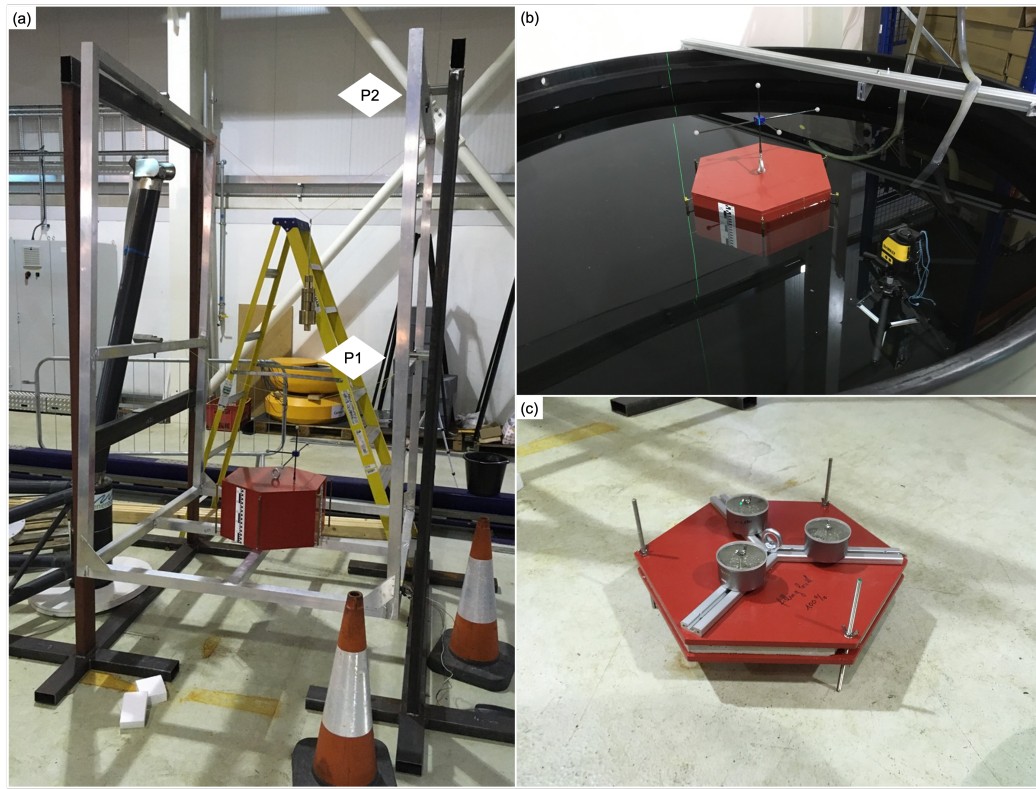

**Figure 2.** Preparation before the deployment in the tank: (**a**) Balanced swing at the pivot axis P2. (**b**) model in the ballast tank to check the correct mass distribution and ballasting. (**c**) example inlet to reach the requested mass properties of the floating device.

**Table 6.** Analysis of the centre of gravity (CG) for the three variations $D1$ to $D3$ for both pivot axes $P1$ and $P2$ (Figure 2a) in relation to the theoretical value–all CG values measured from the outside bottom of the floating structure.

| | | Measured | | | Difference (Value-Theo) | | | Difference/Theo | | |
|---|---|---|---|---|---|---|---|---|---|---|
| | Theo [mm] | P1 [mm] | P2 [mm] | (P1 + P2)/2 [mm] | P1 [mm] | P2 [mm] | (P1 + P2)/2 [mm] | P1 [%] | P2 [%] | (P1 + P2)/2 [%] |
| Conf $D1$ | 119.8 | 122.9 | 110.3 | 116.6 | 3.0 | −9.6 | −3.3 | 2.54% | −7.98% | −2.72% |
| Conf $D2$ | 113.0 | 110.0 | 111.3 | 110.7 | −3.0 | −1.7 | −2.3 | −2.64% | −1.48% | −2.06% |
| Conf $D3$ | 120.8 | 123.6 | 116.4 | 120.0 | 2.8 | −4.4 | −0.8 | 2.33% | −3.66% | −0.67% |

**Table 7.** Analysis of the moment of inertia (MI) for the three variations *D1* to *D3* for both pivot axes *P1* and *P2* (Figure 2a) in relation to the theoretical value–all MI values are referenced to the water level of each individual draft and the $I_{Mod+Swing}$ represent the MI for the investigated model including the swing–all values in [kg m$^2$].

| | | Measured | | | Difference (Value-Theo) | | | Difference/Theo | | | $I_{Mod+Swing,WL}$ |
|---|---|---|---|---|---|---|---|---|---|---|---|
| | Theo | P1 | P2 | (P1 + P2)/2 | P1 | P2 | (P1 + P2)/2 | P1 | P2 | (P1 + P2)/2 | |
| Conf *D1* | 1.789 | 1.482 | 1.079 | 1.280 | −0.307 | −0.402 | −0.095 | −17.19% | −22.49% | −5.31% | 57.824 |
| Conf *D2* | 1.111 | 1.047 | 0.773 | 0.910 | −0.065 | −0.274 | −0.209 | −5.82% | −24.62% | −18.80% | 60.523 |
| Conf *D3* | 0.780 | 0.790 | 0.245 | 0.518 | 0.010 | −0.545 | −0.555 | 1.32% | −69.88% | −71.20% | 63.762 |

### 3.2. Mooring Natural Frequency

As presented in Section 2, two different mooring concepts were investigated. For the version *M1* two mooring lines were connected at one point at each side of the floating structures. This is similar to the set-up used for the experiments with the water filled floating cylinder presented in Gabl et al. [30] and introduces reduced restriction for pitch rotation. Each of the four mooring lines were connected to a separate corner of the prism, which represents a more realistic mooring system. Independently of the variation, all connections points were set to be at the still water surface for both configurations.

Figure 3 shows the results of the analysis of the natural frequency of the four options. The model was moved or angled up to a certain point and the restraining behaviour was recorded with the motion capturing system. This damped oscillation was analysed to calculate the corresponding frequency. Each degree of freedom (DoF) was separately investigated with multiple repetitions. The motions aligned with still water surface, namely surge and sway (*x* and *y*-direction) are very small (under 0.05 Hz). In these DoF, the material properties of the hollow elastic was dominant similar to Gabl et al. [29]. Due to the smaller wave amplitudes the station keeping under waves is better than seen in Gabl et al. [29,30].

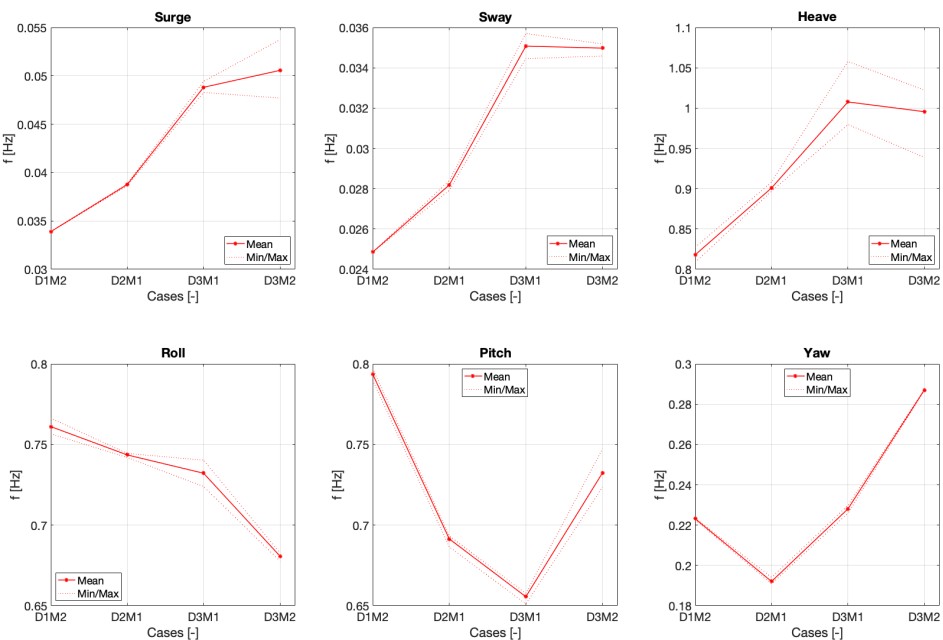

**Figure 3.** Analysis of the natural frequencies of the four different cases including all three draft options *D1–D3* for the two different mooring configurations *M1* and *M2* shown in Figure 1.

The natural frequencies of both mooring options were tested for the smallest draft *D3*. Hence, the comparison of *D3M1* and *D3M2* in Figure 3 provides the direct comparison between both configurations. As expected, the frequency in heave (*z*-direction) was very similar and dominated by the properties of the model. A smaller draft implies a smaller total weight (Table 1), which results in a faster oscillation in the vertical direction. The biggest difference between both options can be found in roll and pitch. *M1* introduces

less restriction in the pitch direction hence the two attachments points connect in a rotation axis parallel to the *y*-axis (Figure 1). The oscillation is slowed down by spreading the connections points hence the mooring lines attenuate the oscillations in the pitch direction. The different lever arms for both options might be the main cause of the differences in the roll direction, but this would have to be tested separately. An increased frequency with a larger draft could also be observed for the cylinder model [29]. The yaw response is comparable low and smaller as the minimum regular wave frequency (Table 3).

### 3.3. Waves Amplitudes and Heights

The first step of the additional analysis is to investigate the measured free surface elevation recorded by the wave gauges (WG). The values $a_W$ (regular waves) and $H_{m0,W}$ (white noise (WN) and JONSWAP irregular waves) represents the requested wave amplitudes and height from the wave makers. This is the input for the transfer function of the tank, which translates the request into actual wave maker motions. It is well known [30,37,48], that the currently used transfer function in FloWave in most cases underestimates the wave amplitude/height and this can be corrected with an adaptation of the $a_W$ or $H_{m0,W}$ to reach the target value. The requested and provided wave frequency is very accurate and a high repeatability could be proven. Such a correction was not conducted for the presented study hence the wave were adapted based on the results of the investigation, which resulted in an expanding wave case seen in Tables 3 and 5.

Figure 4 presents the comparison of the three different model configurations as well as the empty tank testing. The influence of the mooring system on the wave field is very small and hence in all cases only the mooring $M1$ is presented. Nevertheless, the full data is available as described in Section 4. The specific wave input values are summarised in Table 3. This analysis show the wave amplitude $a$ fitted to the measurements of individual or summarised WG. The time was limited to the repeat time of 128 s. In the left graph the actual values are presented and in the right one the differences to the requested wave amplitude $a_W$. The results for the regular waves are shown in relation to the wave frequency $f$. This systematic is similarly used for the white noise test in Figure 5 and for the JONSWAP irregular waves in Figure 6, but for those waves the wave frequency is replaced by the case number.

Regular waves were tested in two wave directions and are separately analysed in Figure 4. The WG are aligned with the *x*-axis, which is identical to the 90° wave direction in the tank definition (Figure 1). By changing the wave direction to the 0°, all WG were in one line and measured the same wave. The comparison of the left and right column of Figure 4 shows this change of the wave direction. This comparison shows a very similar picture of a comparable good agreement in the lower frequencies and a larger spread for higher frequencies for both wave directions. Consequently, the influence of the different wave direction is very small. The investigated regular waves had a requested wave amplitude of 5 mm, which is relatively small even for a very good accuracy of the WG, which are typically smaller than 1 mm (Section 2.2). A larger amplitude would have been easier to generate but due to the relatively small freeboard of the model configuration $D1$, this value was chosen. The primary WG for the representation of the wave field has to be chosen carefully including a larger relative variability.

The analysis for the WN and the JONSWAP irregular wave tests were conducted similar to the regular waves. Figure 5 and 6 show the results for the measured wave heights $H_{m0}$ in relation to the case number. Depending on the specific wave heights the difference between the requested and measured vary but with a common tendency to measure a slightly smaller height than it was requested. This is typical for the current FloWave transfer function and could be corrected by an adapted gain factor for the input $H_{m0,W}$. The right graph with the differences uses the requested value as well as the empty tank result. The latter allows to identify the influence of the reflection caused by the model. The largest differences can be found for the variation $D1$, which had the biggest draft. Those values are significantly smaller for the other variations with a reduced draft.

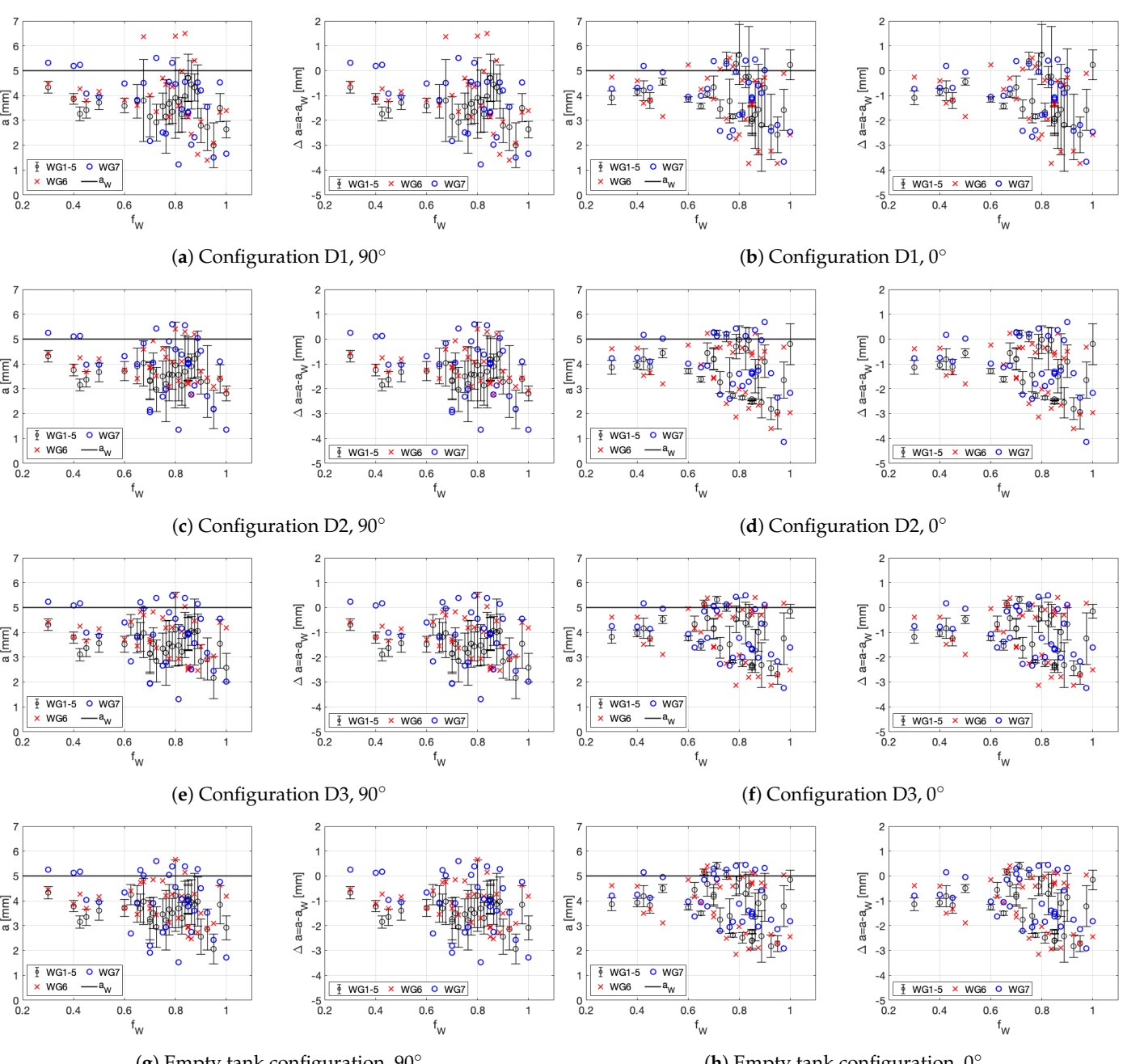

**Figure 4.** Regular waves-Analyses of the wave gauges (WG) for the three different configurations of the model (*D*1–*D*3) for the mooring system *M*1 and the empty tank testing without a model in relation to the wave frequeny $f_W$–separated by the wave direction along the WG (90° in the tank configuration) and orthogonal to them (0°) and analysed in relation to the requested wave amplitude $a_W$.

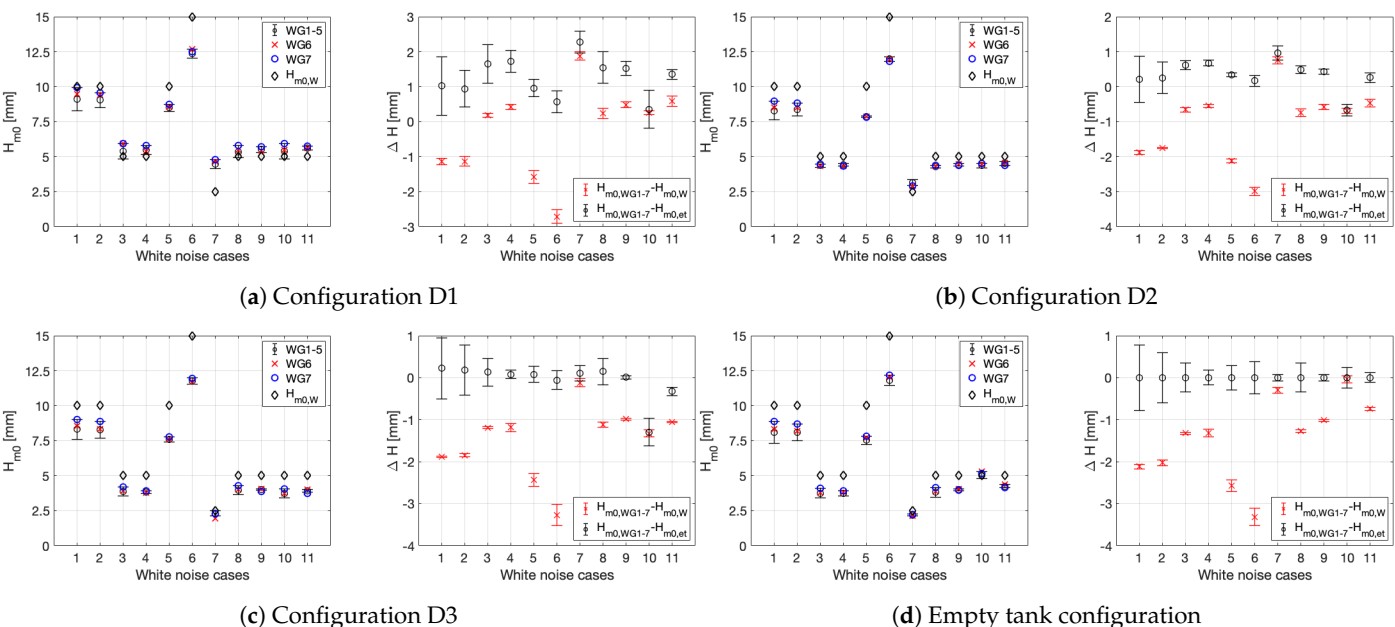

**Figure 5.** White noise-Analyses of the wave gauges (WG) for the three different configurations of the model (*D*1–*D*3) for the mooring system *M*1 and the empty tank testing without a model for the 11 cases presented in Table 4.

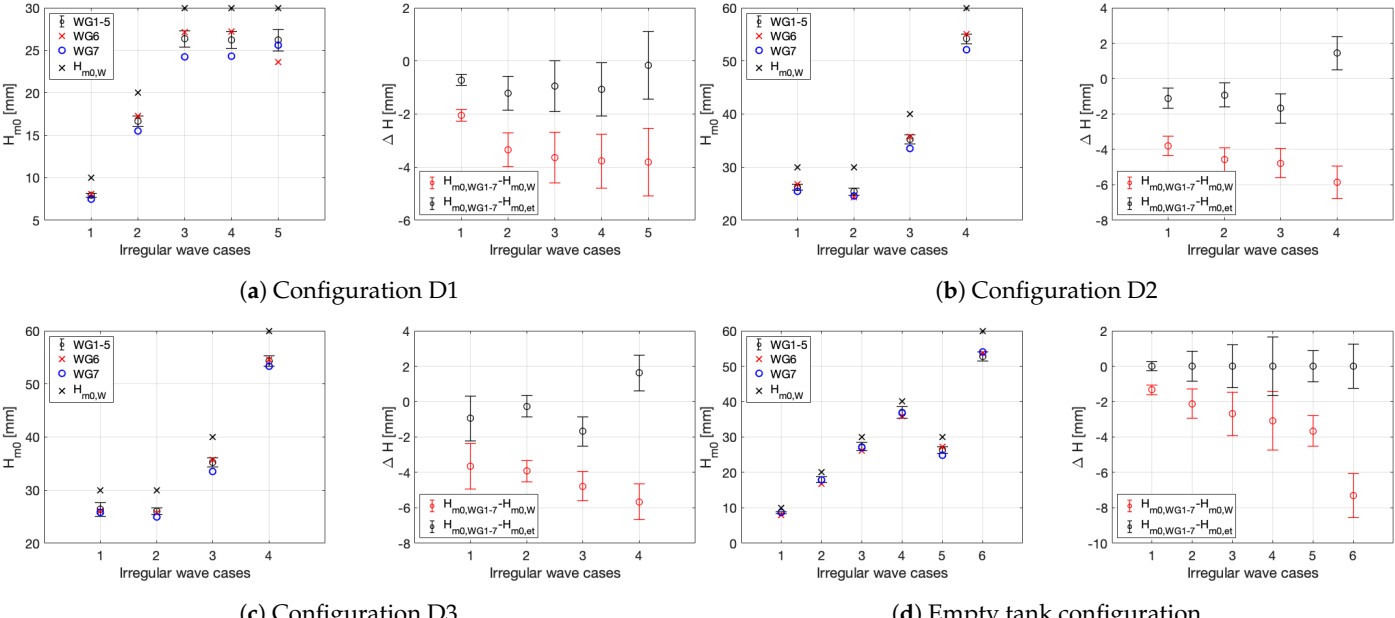

**Figure 6.** JONSWAP irregular waves—Analyses of the wave gauges (WG) for the three different configurations of the model (*D*1–*D*3) for the mooring system *M*1 and the empty tank testing without a model for the different cases presented in Table 5.

## 4. Dataset Description

The available data is grouped for the individual drafts *D*1–*D*3 (Table 1) and the empty tank *ET*. Table 8 presents the content of each individual file, which includes the wave gauges *WG*1 to *WG*7 (location provided in Table 2) and the trigger signal in all cases. The latter is a recording of the tank trigger, which was used for the synchronisation of the motion capturing system (MoCAP). The specific value is not calibrated and only the time of the raising and falling signal is important, which indicates starting and stopping of the wave makers. It is included to allow a reproduction the time series covering the run and repeat time based on the full capturing time. Each of those time windows of the same wave is stored in a separate folder as part of the provided dataset [4].

**Table 8.** Available data for each file–the empty tank files only include the column 1 to 8.

| Name | WG1 | WG2 | WG3 | WG4 | WG5 | WG6 | WG7 | Trigger |
|------|-----|-----|-----|-----|-----|-----|-----|---------|
| Unit | [mm] | [mm] | [mm] | [mm] | [mm] | [mm] | [mm] | [-] |
| Column | 1 | 2 | 3 | 4 | 5 | 6 | 7 | 8 |
| Name | X | Y | Z | RZ | RY | RX | Residual | |
| Unit | [mm] | [mm] | [mm] | [deg] | [deg] | [deg] | [mm] | |
| Column | 9 | 10 | 11 | 12 | 13 | 14 | 15 | |

The *CaptureTime* folder provides the files for the full recorded time. This includes the full time when the wave makers were active, which is provided in the *RunTime* folder, and further seconds to document the decreasing motion response. The time window for the files in the *RepTime* folder is further limited to the specific repeat time of the individual waves ending when the wave makers stop. This allows for a fully developed motion response, which has to be verified individually. Depending on the exact purpose a further limitation is advisable.

For the cases with a model in the tank, the motion in the main direction $X$, $Y$ and $Z$ are provided in the tank coordinate system (Figure 1) as well as the other three rotational DoF. For example, $RZ$ includes the rotation around the vertical $z$-axis, which is independently of the wave direction the yaw angle. Pitch and roll is depending on the wave direction.

Three different types of waves were investigated and made available: (a) regular waves *Reg* (detailed input values are provided in Table 3), (b) white noise *WN* (Table 4) and (c) JONSWAP irregular waves *Irr* (Table 5). The name of the individual file is a combination based on the following rules:

D*[draft ID]*_ M1 *or* M2_ 00deg *or* 90deg_ Reg *[wave ID]*
D*[draft ID]*_ M1 *or* M2_ WN *[wave ID]*
D*[draft ID]*_ M1_ Irr *[wave ID]*
ET_ 00deg_ Reg *or* 90deg_ Reg *or* WN *or* Irr *[wave ID]*

The *[draft ID]* indicates the different model configurations. In combination with the mooring configuration, either *M*1 or *M*2, the set-up is defined. With the *[wave ID]* the individual wave is marked and same IDs represent the similar wave inputs. The regular waves are further separated by the two different wave directions. Irregular waves were only tested for the mooring configuration *M*1. Table 9 summarises the available videos for those test, which was done based on two perspectives. The detailed view shown in Figure 7a documents the run up on the side based on a video camera (FLIR BFLY-PGE-23S6C-C) and an additional overview is provided based on GoPro footage. All data is available via Edinburgh DataShare [4].

**Table 9.** Overview of the available videos for the JONSWAP irregular waves–input variables presented in Table 5 and the difference between detail and overview is shown in Figure 7—Empty tank conditions were not documented with a specific video.

| | Conf *D1* | | Conf *D2* | | Conf *D3* | |
|------|--------|----------|--------|----------|--------|----------|
| | Detail | Overview | Detail | Overview | Detail | Overview |
| Irr1 | | | x | x | x | x |
| Irr2 | x | | x | x | x | x |
| Irr3 | x | | x | x | | x |
| Irr4 | x | x | x | x | x | x |
| Irr5 | x | x | | | | |

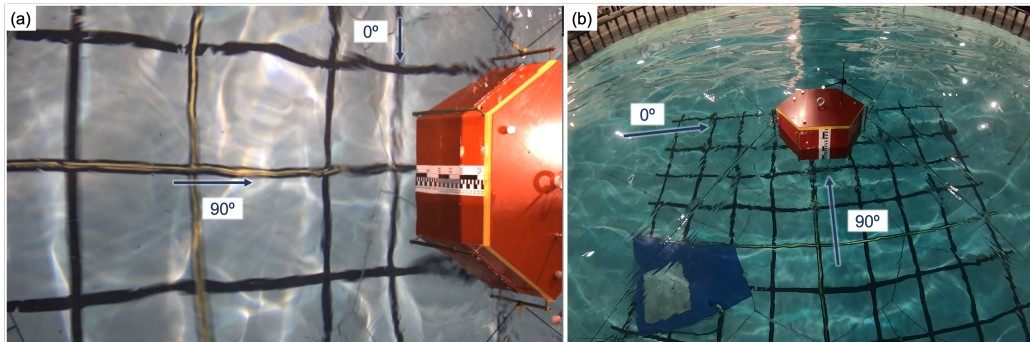

**Figure 7.** Overview of the videos: (**a**) detailed video with precise frame capturing. (**b**) Overview captured with a GoPro.

**Author Contributions:** Conceptualization and measurement R.G. and R.K.; methodology R.G., R.K., T.D. and D.M.I., formal analysis, R.G., R.K. and T.D.; writing—original draft preparation, R.G. and T.D.; writing—review and editing, R.K. and D.M.I. All authors have read and agreed to the published version of the manuscript.

**Funding:** This work was supported by the Austrian Science Fund (FWF) under Grant J3918 (R.G.) and a K-Regio project BEQs-Buoyant Energy Quarters, which is part of the European Regional Development Fund (ERDF).

**Institutional Review Board Statement:** Not applicable.

**Informed Consent Statement:** Not applicable.

**Data Availability Statement:** The data presented in this study are openly available: https://doi.org/10.7488/ds/3125.

**Conflicts of Interest:** The authors declare no conflict of interest.

## Abbreviations

The following abbreviations are used in this manuscript:

| | |
|---|---|
| $a$ | amplitude waves (mm) |
| BEQ | Buoyant Energy Quarters |
| $a_W$ | amplitude waves (mm) requested from the wave makers |
| C | cases |
| CG | centre of gravity |
| DoF | degree of freedom |
| $ET$ | empty tank |
| $h$ | draft of the floating structure (mm) |
| $H$ | total height of the prism (mm) |
| $H_{m0}$ | wave height irregular waves (mm) |
| $H_{m0,W}$ | wave height irregular waves (mm) requested from the wave makers |
| MI | moment of inertia |
| MoCAP | motion capturing system |
| RAO | response amplitude operator |
| $s$ | side length of the hexagon |
| VLFS | Very large floating structure |
| WEC | wave energy converter |
| WG | wave gauge |
| WL | water level |

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
