# Peer review of "Experimental Data of a Hexagonal Floating Structure under Waves"

_data, 2021_

Round 1

Reviewer 1 Report

Some comments are as follows:

  1. The description from L85 to L136 is recommended to add illustrations for the clarity. The photos shown in the manuscript are difficult for readers to understand the settings of the experiment, for instance, the angles of cables, etc.
  2. Please add the information of wave heights in Table 3.
  3. In Table 5, the smallest Hm0 is 1 cm, and the depth of the flume is about 2 m. Doesn't it cause issues/errors for the experiments?

Author Response

We thank the reviewer for the comments and suggestions. The attached file includes a response to all three reviewers. Thank you!

Reviewer 2 Report

The paper presents data from an experimental test campaign of a floating moored structure. My comments are listed below:

Introduction:

  • If possible, I believe it could be interesting to present more about the Buoyant Energy Quarters.
  • Other studies have also done validation test on large floating wave energy structures, with focus on frequency domain and mooring solvers:
    • Thomsen, J. B., Ferri, F., & Kofoed, J. P. (2016). Experimental testing of moorings for large floating wave energy converters. I C. G. Soares (red.), Progress in Renewable Energies Offshore: proceedings of RENEW 2016, 2nd international conference on renewable energies offshore (s. 703-710). CRC Press. https://doi.org/10.1201/9781315229256-83
    • Thomsen, J. B., Ferri, F., & Kofoed, J. P. (2017). Validation of a Tool for the Initial Dynamic Design of Mooring Systems for Large Floating Wave Energy Converters. Journal of Marine Science and Engineering, 5(4), [45]. https://doi.org/10.3390/jmse5040045

Experimental Set-up:

  • Section 2.1: A drawing of the model would be useful.
    • How much reflection was present in the tank? Both in the wave direction and in the sideway direction?
    • I believe it would be useful with a better description of the mooring system. The two configurations are not clear to me.
  • Section 2.3:
    • Line 161: Delete one of the “and”.
    • Was only one wave height tested? And what was the wave height (I think it should be presented in the text and not only in the caption of Table 3)? RAOs can be non-linear due to the shape of the structure and the mooring system. Meaning that two wave heights with the same frequency can provide different response even when normalized according to the wave height. Was this tested? If not, I think it should be addressed and the wave height presented.
    • Table 4: I think the columns need more explanation. What is meant by capture, run time and repeat? How was the length of the series chosen?
    • Table 5: What does Irr1-6 mean? What was the length of this time series and why was it chosen (present in text)?
  • Section 3.2: How was the natural period measured?

Author Response

(The authors gave the same response as above.)

Reviewer 3 Report

This is a nice paper, which in my opinion can be published after the minor revisions listed below:

  • Facility characteristics, wave directions and geometrical variables of the object should be highlighted via ad hoc sketches; this would raise the readability of the paper.
  • On lines 92: “It is assumed that all other direction lead to an additional rotation around the z-axis due to the used very soft mooring system. This would not be ideal for a validations experiment hence another uncertainty is introduced in the comparison” This phrase is unclear and must be re-written;
  • On row 96: “the perpendicular incoming wave” . Does this mean, perpendicular to the object’s side? Please clarify.
  • Rows 126-127. Sampling frequency of wave gauges?
  • Row 156. White Noise waves are in fact Irregular waves. What is the point of distinguishing between “White noise waves” and “Irregular waves”? I suggest turning the nomenclature into “b) White Noise irregular waves (WN) and c)JONSWAP irregular waves (J)

Author Response

(The authors gave the same response as above.)
